# Biotransformation of Modified Benzylisoquinoline Alkaloids: Boldine and Berberine and In Silico Molecular Docking Studies of Metabolites on Telomerase and Human Protein Tyrosine Phosphatase 1B

**DOI:** 10.3390/ph15101195

**Published:** 2022-09-28

**Authors:** Duaa Eliwa, Abdel-Rahim S. Ibrahim, Amal Kabbash, Mona El-Aasr, Michał Tomczyk, Yousef A. Bin Jardan, Gaber El-Saber Batiha, Amany E. Ragab

**Affiliations:** 1Department of Pharmacognosy, Faculty of Pharmacy, Tanta University, Tanta 31527, Egypt; 2Department of Pharmacognosy, Medical University of Białystok, ul. Mickiewicza 2a, 15-230 Białystok, Poland; 3Department of Pharmaceutics, College of Pharmacy, King Saud University, Riyadh P.O. Box 2455, Saudi Arabia; 4Department of Pharmacology and Therapeutics, Faculty of Veterinary Medicine, Damanhour University, Damanhour 22511, Egypt

**Keywords:** biotransformation, boldine, berberine, *Cunninghamella elegans* NRRL 2310, *Rhodotorula rubra* NRRL y1592, Penicillium chrysogeneum ATCC 10002, *Cunninghamella blackesleeana* NRRL 1369, *Cunninghamella blackesleeana* MR198, telomerase, human protein tyrosine phosphates 1B

## Abstract

Natural nitrogen heterocycles biotransformation has been extensively used to prepare synthetic drugs and explore the fate of therapeutic agents inside the body. Herein, the ability of filamentous fungi to biotransform boldine and berberine was investigated. Docking simulation studies of boldine, berberine and their metabolites on the target enzymes: telomerase (TERT) and human protein tyrosine phosphatase 1B (PTP-1B) were also performed to investigate the anticancer and antidiabetic potentials of compounds *in silico*. The biotransformation of boldine and berberine with *Cunninghamella elegans* NRRL 2310, *Rhodotorula rubra* NRRL y1592, *Penicillium chrysogeneum* ATCC 10002, *Cunninghamella blackesleeana* MR198 and *Cunninghamella blackesleeana* NRRL 1369 via demethylation, N- oxidation, glucosidation, oxidation and hydroxylation reactions produced seven metabolites, namely: 1,10-didesmethyl-boldine (**1**), laurolitsine (**2**), 1,10-didesmethyl-norboldine (**3**), boldine-9-*O*-*β*-D-glucoside (**4**), tridesmethyl berberine (**5**), demethylene berberine (**6**), and lambertine (**7**). Primarily, the structures of the metabolites were established by one-dimensional (1D) and two-dimensional (2D) nuclear magnetic resonance (NMR) analyses and mass spectrometry. In silico molecular docking simulation of the metabolites of boldine and berberine to the proteins TERT and PTP-1B, respectively, revealed good binding MolDock scores comparable to boldine and berberine and favorable interactions with the catalytic sites of the proteins. In conclusion, this study presented promising biologically prepared nitrogen scaffolds (isoquinolines) of boldine and berberine.

## 1. Introduction

Biotransformation is a process by which organic compounds are converted to closely related derivatives to counteract the toxicity of the chemical substances. This process is achieved by microorganisms such as bacteria and fungi, or by activities of various enzymes [1]. Alkaloids have diverse biological and therapeutic activities ranging from useful medicines to potent toxins. Since many important drugs are derived from such natural compounds, there is much interest in their biotransformation to provide new compounds or intermediates for the synthesis of new or improved drugs [2]. Boldine and berberine are both modified benzylisoquinoline alkaloids belonging to aporphine and protoberberine types, respectively, and exhibit diverse biological activities. Boldine is a free radical scavenger implicated in the treatment of various diseases [3]. It has a protective effect against estrogen deficiency (osteoporosis) [4]. Boldine can also counteract endothelial dysfunction associated with hypertension and diabetes mellitus [5]. Furthermore, boldine has gastroprotective effect and is useful in gastric ulcer treatment [6]. As an effective antioxidant, boldine can act as anti-apoptotic agent and protect the liver from damage induced by methylprednisolone [7]. Boldine showed anti-inflammatory activity in Guinea pig Paw edema and reduced hyperthermia in rabbits due to bacterial pyrogens and inhibited prostaglandin biosynthesis [8]. Boldine increased eNOS in penile tissue and improved erectile functions in rats [9]. Recent studies have revealed that berberine exhibits anti-tumor properties against leukemia, lung cancer, cervical cancer, and other cancers [10].

Boldine also inhibits the enzyme telomerase, which is a target enzyme for anticancer drug research. Berberine, on the other hand, has also a protective effect against five metabolic disorders namely type-2 diabetes, obesity, non-alcoholic fatty liver, hyperlipidemia, and gout. These actions are brought about by increasing insulin secretion, improving insulin resistance, inhibiting adipose tissue fibrosis, improving liver steatosis, and improving gut microbiome [11]. Berberine also has antidiabetic effect by stimulating insulin secretion and modulating lipids [12]. In a recent study, berberine was found to have antidiarrheal effect in case of irritable bowel syndrome by inhibiting neurotransmission in colonic smooth muscle [13]. Due to the structural similarity of berberine and papaverine, it also inhibits human protein tyrosine phosphatase 1B (PTP-1B) [14] which plays a major rule in the development of diseases associated with insulin resistance such as obesity and diabetes as it is involved in the dephosphorylation of tyrosine residues of the insulin receptors [15]. 

Preparative scale resolution by *Fusarium solani* and stereospecific oxidation of (*R*, *S*) glaucine, an aporphine alkaloid related to boldine by *Aspergillus flavipes* was investigated [16]. *Penicillium roqueforti* NRRL 849 transformed boldine into boldine *N*-oxide [17]. The direct interaction of berberine and gut microbiota was reviewed in view of low bioavailability of oral berberine and to study altered pharmacological effects displayed by berberine metabolites [18]. Metabolites of berberine by gut microbiota were reviewed [19]. Gut microbiota was found to biotransform berberine to absorbable metabolites (dihydroberberine) [20]. Four metabolites of berberine by cytochrome P450 isoenzyme were identified using LC/tandem mass spectroscopy [21]. Microbial transformation studies of berberine by *Curvularia lunata* was performed but no metabolites were isolated in spite of almost complete consumption of the substrate [22]. Berberine 7-*N*-oxide was isolated as metabolite of berberine by the endophytic fungus *Coelomycetes AFKR-1* obtained from *Archangelisia flava* (Menispermaceae) [23]. The microbial transformation studies of boldine and berberine were found to be scarce, which prompted us to undertake this investigation. Herein, we investigated the microbial transformation of boldine and berberine using a selection of filamentous fungi. Furthermore, molecular docking simulation studies were performed to investigate potential anticancer and antidiabetic metabolites related to boldine and berberine, respectively.

## 2. Results and Discussion

### 2.1. The Biotransformation Products Identification

The initial screening of isoquinoline alkaloid substrates (boldine and berberine) used 50 filamentous fungal strains from different classes. Four metabolites of boldine were isolated from the biotransformation broth and freeze–dried mycelia of Cunninghamella blackesleeana NRRL 1369, Penicillium chrysogeneum ATCC 10002 and Cunninghamella blackesleeana MR 198. Three metabolites of berberine were isolated from the biotransformation broth and freeze–dried mycelia of Cunninghamella elegans NRRL 2310, Cunninghamella blackesleeana NRRL 1369 and Penicillium chrysogeneum ATCC 10002. On the basis of their mass and NMR spectroscopic data, the metabolites were identified. 

The ^1^H NMR data of boldine metabolite-**1** (Table 1, Appendix A) displayed the lack of two singlet signals at δ_H_ 3.55 and 3.77 each integrating for three protons (O-1 and O-10 methyl groups) which are present in boldine. The ^13^C NMR of 1 (Table 1, Appendix A) showed 17 peaks which correspond to 17 different carbon atoms less than boldine by two carbon atoms and disappearance of two methyl group signals at δ_C_ 59.2 and 55.7 which are present in boldine. Other ^13^C NMR data showed resemblance to those of boldine. DEPT-135 and APT spectra of **1** (Appendix A) proved the presence of three methylene carbons (attributed to C-4, C-5 and C-7), one methyl carbon (N-CH_3_) and four methine carbons (C-3, C-8, C-11 and C-6a). IR spectrum (Appendix A) showed OH group at 3441 cm^−^^1^. A pseudo-molecular ion peak [M+H]^+^ present in (+) ESI-MS spectrum (Appendix A) at *m*/*z* 300 presumably corresponding to a molecular formula of C_17_H_17_O_4_N of which is 28 Da less than that of boldine. Based on boldine metabolite **1** spectral data, it was elucidated as 1,10-didesmethyl-boldine [24] (Figure 1).

The ^1^H NMR data of boldine metabolite-**2** (Table 1, Appendix A) displayed the lack of a singlet signal at δ_H_ 2.38 integrating for three protons which is present in boldine. The ^13^C NMR of 2 (Table 1, Appendix A) showed 18 peaks corresponding to 18 different carbon atoms which is less than boldine by one carbon atom and disappearance of one methyl group signal at δ_C_ 43.7 present in boldine. DEPT-135 and APT spectra of 2 (Appendix A) proved the existence of three methylene carbon (attributed to C-4, C-5 and C-7), two methyl carbons (O-1 and O-10 methyl groups) and four methine carbons (C-3, C-8, C-11 and C-6a). IR spectrum (Appendix A) showed OH group signal at 3429 cm^−1^. A pseudo-molecular ion peak [M+H]+ at *m*/*z* 314 was detected in the (+) ESI-MS spectra (Appendix A), which is presumably corresponding to a molecular formula of C_18_H_19_O_4_N which is 14 Da less than that of boldine. Based on boldine metabolite-**2** spectral data, it was elucidated as nor-boldine or laurolitsine or boldine N-demethyl [25] (Figure 1).

The ^1^H NMR data of boldine metabolite-**3** (Table 1, Appendix A) showed disappearance of three singlet signals at δ_H_ 2.38, 3.55 and 3.77 each integrating for three protons (N-CH_3_, O-1 and O-10 methyl groups) which are present in boldine resulting from the triple demethylation reactions. ^13^C NMR of **3** (Table 1, Appendix A) showed 16 peaks which correspond to 16 different carbon atoms less than boldine by three carbon atoms and disappearance of three methyl groups signal at δ_C_ 43.7, 55.7 and 59.2 present in boldine. DEPT-135 and APT spectra of **3** (Appendix A) proved the existence of three methylene carbon (attributed to C-4, C-5 and C-7) and four methine carbons (C-3, C-8, C-11 and C-6a). IR spectrum (Appendix A) showed OH group band at 3483 cm^−1^. A pseudo-molecular ion peak [M+H]^+^ at *m*/*z* 286 was present in (+) ESI-MS spectrum (Appendix A) presumably corresponding to a molecular formula of C_16_H_15_O_4_N of which 42 Da less than that of boldine. Based on boldine metabolite-**3** spectral data, it was elucidated as 1,10-didesmethyl-norboldine [24] (Figure 1).

(+) ESI-MS spectrum of boldine metabolite-**4** (Appendix A) revealed the presence of a pseudo-molecular ion peak [M+H]^+^ at *m*/*z* 490, which is probably corresponding to a molecular formula of C_25_H_31_O_9_N. IR spectrum (Appendix A) showed OH group frequency at 3392 cm^−1^ and a conjugated carbonyl (C=O) sharp band at 1581 cm^−1^. Other bands include phenolic C‒O stretching at 1262 cm^−1^ and alcoholic C‒O stretching at 1141 and 1037 cm^−1^. The ^1^H NMR data of 4 (Table 1, Appendix A) showed a singlet of one proton intensity at δ_H_ 6.52 belongs to proton on C-3. Multiplet at δ_H_ 2.51–2.65 integrating for three protons belongs to two protons on C-4 and one proton on C-6a. Another multiplet at 2.96–3.20 ppm integrating for four protons belongs to two protons on C-5 and two protons on C-7. The remaining two signals obviously belonged to sugar part with one six protons multiplet at δ_H_ 4.00–4.26 signifying sugar protons and the other one proton doublet of the anomeric proton (Glu-H-1`) resonating at δ_H_ 5.05. Based on anomeric proton chemical shift which was located downfield from the bulk of sugar protons and the large coupling constant (*J* = 7.2 Hz), it can be concluded that the aglycone binds to the sugar through a *β*-glycosidic linkage. Examination of ^13^C NMR of boldine metabolite-**4** (Table 1, Appendix A) showed 25 peaks due to 25 different carbon atoms which corresponding to 19 carbon atoms of boldine and six carbons of glucose. DEPT-135 and APT spectra of boldine metabolite-**4** (Appendix A) proved the existence of four methylene carbons (attributed to C-1`, C-4, C-5 and C-7), three methyl carbons (N-CH_3_, O-1- CH_3_ and O-10- CH_3_) and nine methine carbons (C-3, C-8, C-11, C-2`, C-3`, C-4`, C-5`, C-6` and C-6a). The attachment of the sugar to C-9 was confirmed by HMBC and HSQC, which showed correlation of the anomeric proton to C-9 (Appendix A). Acid hydrolysis of boldine metabolite-**4** afforded glucose (Co-TLC) and boldine aglycone (Co-TLC). The structure boldine glucoside was thus proposed for this compound. Based on boldine metabolite-**4** spectral data, it was elucidated as boldine-9-*O*-*β*-D-glucoside (Figure 1). This is the first isolation of boldine-9-*O*-*β*-D-glucoside.

The ^1^H NMR data of berberine metabolite-**5** compared to that reported for berberine [26] (Table 2, Appendix A) displayed the lack of three singlet signals at δ_H_ 4.21, 4.11 and 6.11 integrating for methoxy group protons on C-9, C-10 and C-2,3 methylene group, respectively, which are present in berberine and indicate three demethylation reactions. The ^13^C NMR of berberine metabolite-**5** (Table 2, Appendix A) showed 17 peaks which correspond to 17 different carbon atoms and the lack of two methyl group signals at δ_C_ 57.1 and 62.0 and one methylene carbon signal at δ_C_ 102.1 present in berberine. Other ^13^C NMR data showed resemblance to those of berberine. DEPT-135 spectrum of 5 (Appendix A) proved the existence of two methylene carbons (attributed to C-5 and C-6) and six methine carbons (C-1, C-4, C-8, C-11, C-12 and C-13). IR spectrum (Appendix A) showed the appearance of OH group signal at 3428 cm^−1^ which is not present in berberine spectrum as previously reported. (+) ESI-MS spectrum (Appendix A) showed the existence of a pseudo-molecular ion peak [M+H]^+^ at *m*/*z* 297 presumably corresponding to a molecular formula of C_17_H_14_O_4_N which is 40 Da lower than that of berberine resulting from three demethylation reactions. Based on berberine metabolite-**5** spectral data, it was elucidated as tridemethyl berberine [27] (Figure 2).

The ^1^H NMR data of the berberine metabolite-**6** compared to that reported about berberine [25] (Table 2, Appendix A) showed two sharp three proton singlets at δ_H_ 4.10 and 4.07 assigned for 9 and 10 methoxy moieties, respectively. It also displayed the lack of one singlet signal at δ_H_ 6.11 integrating for two protons corresponding to C-2,3 methylene protons which is present in berberine, suggesting a demethylation reaction. The ^13^C NMR of 6 (Table 2, Appendix A) showed 19 peaks which correspond to 19 different carbon atoms and the lack of one methylene carbon signal at δ_C_ 102.1 present in berberine. Other ^13^C NMR data showed resemblance to those of berberine. DEPT-135 spectrum of 6 (Appendix A) proved the existence of two methylene carbons (attributed to C-5 and C-6), two methyl carbons (two methoxy groups at C-9 and C-10) and six methine carbons (C-1, C-4, C-8, C-11, C-12 and C-13). IR spectrum (Appendix A) showed the appearance of OH band signal at 3451 cm^−1^ which is not present in berberine spectrum as previously reported. (+) ESI-MS spectrum (Appendix A) showed the presence of a pseudo-molecular ion peak [M+H]^+^ at *m*/*z* 325 presumably corresponding to a molecular formula of C_19_H_18_O_4_N of which is 12 Da lower than that of berberine (loss of a methylene group). Based on berberine metabolite-**6** spectral data, it was elucidated as demethylene berberine (DMB) [27] (Figure 2).

The ^1^H NMR data of the berberine metabolite-**7** compared to that reported about berberine [25] (Table 2, Appendix A) displayed a lack of a singlet signal of one proton intensity of H-8 at δ_H_ 9.77 present in berberine and appearance of a new singlet signal of two proton at δ_H_ 4.30 ppm which can be ascribed to a reduction reaction of bond between C-8 and the nitrogen atom. The ^13^C NMR of 7 (Table 2, Appendix A) showed 20 peaks corresponding to 20 different carbon atoms which is the same number of carbon signals present in berberine. There is a significant shielding of C-8 signal (by 94 ppm) due to the reduction reaction. Other ^13^C NMR data showed resemblance to those of berberine. DEPT-135 spectrum of berberine metabolite-**7** (Appendix A) proved the existence of four methylene carbons (attributed to C-5, C-6, O-CH_2_-O and C-8), two methyl carbons (two methoxy groups at C-9 and C-10) and five methine carbons (C-1, C-4, C-8, C-11, C-12 and C-13). (+) ESI-MS spectrum (Appendix A) showed the presence of a pseudo-molecular ion peak [M+H]^+^ at *m*/*z* 338 presumably corresponding to a molecular formula of C_20_H_19_O_4_N which one Da higher than that of berberine. Based on berberine metabolite-**7** spectral data, it was elucidated as dihydoroberberine (lambertine) [27] (Figure 2). 

Dealkylation is the most common reaction occurred in boldine and berberine indicating that several monooxygenases with different substrate specificities are involved in this reaction [24] have revealed that boldine shows phase II glucuronidation in rats. Their study demonstrated that treatment of urine samples with β-glucuronidase increased the recovery of boldine three to four-fold. This provided indirect evidence of the extensive formation of boldine glucuronides. Following this, more analysis was detailed through the use of the LC-MS method, which confirmed glucuronide and sulfate conjugates as the major metabolites of boldine in rats. The main metabolites identified using LC–MS were nor-boldine, boldine-9-O-glucuronide, boldine-2-O-glucuronide, boldine-O-sulphate and disulphate, boldine-O-glucuronide-O-sulphate and N-demethyl-boldine-O-sulphate [28]. In our study, we identified boldine-9-O-β-glucoside and nor-boldine as microbial metabolites of boldine. 

In rat studies, Wang et al. investigated plasma profiles of berberine and its metabolites after berberine intravenous administration (4 mg/kg). The major circulating metabolites of berberine were berberrubine, demethyleneberberine and their corresponding glucuronides [27]. Additionally, berberine was converted into dihydroberberine (dhBBR) found only in the feces of rats [20]. In the current study, we identified demethyleneberberine and dihydroberberine as microbial biotransformation products of berberine, which reinforce the concept of microbial models of mammalian drug metabolism. It was found that the demethylation of berberine enhances its intestinal absorption in mice in vivo, which increases the therapeutic potential of berberine [29]. Furthermore, dihydroberberine is considered an active metabolite as it showed up to five times more bioavailable/absorbable and two times longer-lasting than berberine (8 h vs. 4 h) [30]. Overall, these findings further support microbial models of drug metabolism in mammals.

### 2.2. Molecular Docking Investigations

#### 2.2.1. Molecular Docking of Boldine Metabolites to Telomerase Enzyme

Telomerase is a multicomponent enzyme which includes a catalytic telomerase reverse transcriptase (TERT) protein and a noncoding RNA molecule which serves as the template for the synthesis of telomere DNA [31]. Telomerase inhibition results in a sequential telomere shortening that triggers apoptosis or senescence in cancer cells [32]. In an in vitro study [33], boldine was proved to be an inhibitor of TERT. In our study, boldine and its metabolites were docked to the TERT deoxynucleotide binding site as a potential inhibitory target to investigate the potential anticancer activity for the isolated metabolites. The method was validated by redocking of 5-methylcarboxyl-indolyl-2′-deoxyriboside 5′-triphosphate (5-MeCITP), the co-crystalized inhibitor for TERT (PDB: 6E53). The MolDock scores (Table 3) of boldine, boldine metabolites **1**, **2**, **3** and **4** were −120.14, −112.52, −135.54, −119.00, and −151.75, respectively. These results indicated a potential binding of the investigated metabolites at the deoxynucleotide binding site in which boldine and its metabolites occupy the site by displacing the RNA template as 5-MeCITP does. Boldine formed hydrogen bonding interactions with Arg194, Asp343 and Tyr256 which helped to stabilize boldine in the docking cavity (Figure 3). Boldine metabolite **1** showed two hydrogen bonding interactions namely with Asp343 and Gly309, with a one fewer bond than boldine, and a steric interaction with Arg194 and Ile196 which may explain the drop of MolDock score from −120.14 to 112.52. Boldine metabolite **2** exhibited hydrogen bonding with Asp343 and Gly309 in addition to steric interactions with Arg194 and Gly309 which were expected to be of favorable as evidenced by the increase of the MolDock score to −135.53. For boldine metabolite **3**, 3 hydrogen bonds were observed with Asp343, Gly309 and Tyr256 which may explain the small difference in the MolDock score compared to boldine. Boldine metabolite **4** exhibited the highest MolDock score amongst the examined metabolites which is supported by the formation of 5 hydrogen bonds namely between the sugar moiety of boldine metabolite4 and each of the residues Lys372, Lys189, Arg194, Asp245 and Ala255. A previous docking study of boldine to TERT [33] indicated that the conserved catalytic residues include Asp343 and Lys372. Asp343 coordinates with Mg^2+^ which is essential for the catalytic reaction, while Lys372 is the base source required for the condensation reaction of the deoxynucleotide. In our investigation, boldine and boldine metabolites **1**-**3** showed hydrogen binding to Asp343, while boldine metabolite **4** exhibited hydrogen binding to Lys372 which implied that these compounds block the catalytic site of TERT and thus potentially inhibit the telomerase enzyme. The results indicated that the isolated metabolites might exert anticancer activity through the inhibition of telomerase enzyme.

#### 2.2.2. Molecular Docking of Berberine Metabolites to the Protein Tyrosine Phosphatase 1B (PTP-1B)

PTP-1B is considered as an attractive target for type-2 diabetes and obesity treatment [15]. Berberine is a promising antidiabetic alkaloid with evidence on inhibition of PTP-1B. Berberine and its metabolites (**5**–**7**) were docked to PTP-1B enzyme to predict their antidiabetic potential. The method was validated by redocking of 1,2,3,4-tetrahydroisoquinolyl sulfamic acid derivative, an inhibitor co-crysalized with PTP-1B (PDB: 2F70). The MolDock scores of berberine, berberine metabolites **5**, **6**, and **7** were −123.57, −115.91, −115.6 and −106.05 (Table 3). Berberine and berberine metabolites **5** and **6** showed similar hydrogen bonding interaction with water molecules and the residues Arg221, Cys215 and Ser216. Additionally, berberine showed hydrogen bonding interaction with Gly220. Thus, demethylation involving dioxy-methylene ring opening reactions did not have a significant effect on the binding interaction profile. For berberine metabolite **7**, hydrogen bonding interaction was observed with Arg221, Cys215, Gly220 and steric interactions with Ala217, Gln262, Asp181 and Cys215. The results indicated good binding affinity to the protein and thus putative antidiabetic activity (Figure 4).

## 3. Materials and Methods

### 3.1. General Experimental Procedure

1D (^1^H, ^13^C, DEPT 135 and APT) and 2D (HMBC and HSQC) NMR spectra were recorded on a Bruker model AMX 400 NMR spectrometer (Bruker Daltonics GmbH & Co. KG, Fahrenheitstraße 428359, Bremen, Germany) with standard pulse sequences, operating at 400 MHz in ^1^H and 100 MHz in ^13^C. The chemical shift (δ) values were reported in parts per million units (ppm) and tetramethylsilane or known solvent shifts, used as internal chemical shift references. Coupling constants (*J* values) were recorded in Hertz (Hz). Standard pulse sequences were used for HSQC, HMBC, APT and DEPT. Mass Spectrometer Thermo Scientific ISQ Quantum Access MAX triple Quadrupole system, X calibur 2.1 software, Waltham, Massachusetts 02451, USA for ESI-MS. TLC was performed using precoated TLC sheets silica gel G 254 F sheets (E. Merck, Germany). Column chromatography was carried out using silica gel (E. Merck, 70–230 mesh) and Sephadex LH-20 (Sigma-Aldrich chemical Co., St. Louis, MI, USA). All the reagents and solvents used for separation and purification were of analytical grade. The solvent system for TLC analysis of boldine, using silica gel plates was, S_1_: chloroform: methanol: acetic acid (80: 20:2, *v*/*v*). For berberine, using silica gel plates; S_2_: n-butanol-acetic acid-water (BAW) 40:10:50 (*v*/*v*) equilibrated in a separator funnel and the upper organic phase was used. The plates were dried and visualized under UV-light at 254 and 365 nm and sprayed with Dragendorff’s and anisaldehyde/sulfuric acid spray reagents.

### 3.2. Chemicals

Berberine was obtained from Santa Cruse Chemical Company Inc, Dallas, TX, USA. Boldine was obtained from Sigma Aldrich Chemical Company Inc, St. Louis, MI, USA. Identity was confirmed by spectral analysis (MS and NMR).

### 3.3. Microorganisms

Initial screening procedure was conducted as previously reported Eliwa et al. [34]. In this case, 50 microbial cultures, obtained from the American Type Culture collection (ATCC, Rockville, Maryland), Northern Regional Research Laboratories (NRRL, Peoria, IL, USA. The initiation and maintenance of the cultures were conducted as previously reported in Eliwa et al. [32]. On the basis of this screening process, several micro-organisms were found to metabolize the alkaloidal substrates very well with variable efficiencies without optimization. *Penicillium chrysogeneum* ATCC 10002, *Cunninghamella blackesleeana* NRRL 1369 and *Cunninghamella blackesleeana MR 198 were the most efficient microorganisms capable* to perform maximum conversion of boldine into its metabolites. *Cunninghamella elegans* NRRL 2310, *Cunninghamella blackesleeana* NRRL 1369 and *Penicillium chrysogeneum* ATCC 10002, showed different metabolites of berberine without optimization.

To produce a significant amount of metabolites, the same technique was used. The highest yielding microbial strains were chosen to biotransform the two alkaloidal substrates for the large scale fermentation experiments. The liquid medium composition and the extraction of the metabolites were performed as previously reported in Eliwa et al. [34]. In all stage II fermentations, we used 0.5 L flasks containing 100 mL stage II cultures of the microorganism.

### 3.4. Preparative Scale Fermentation and Purification of the Metabolites

Several large scale fermentations were conducted as follows.

#### 3.4.1. Boldine Transformation by *Cunninghamella blackesleeana* NRRL 1369

300 mg substrate/7.5 mL DMF was equally partitioned among 30 stage II culture. Fermentation residue (650 mg) was obtained after 15 days. The residue was dissolved in 2 mL of a methanol–dichloromethane mixture (1:1) and placed on top of silica gel column (70 g, 120 cm × 2.5 cm). The column was eluted by the gradient elution method in which 100% ethyl acetate was initially used, then gradually increasing the polarity by adding methanol by 5% increment up to 80% methanol. Fractions 50–66 afforded impure boldine metabolite **1**. Those fractions were combined and evaporated to dryness (20 mg). Further purification was achieved utilizing the Sephadex LH-20 column (5 g, 60 cm × 1 cm), which was eluted severally with 100% methanol to yield pure boldine metabolite **1** (1,10-didemethyl-boldine), (5 mg, 1.7% yield, *Rf* = 0.55, S1), as amorphous white powder.

#### 3.4.2. Boldine Transformation by *Cunninghamella blackesleeana* MR 198

Boldine (400 mg/10 mL DMF) was equally distributed among 40 stage II flasks. Fermentation residue (850 mg) was obtained after two weeks. The residue was dissolved in 2 mL of a methanol–dichloromethane mixture (1:1) and placed on top of silica gel column (70 g, 120 cm × 2.5 cm). The column was eluted by the gradient elution method using ethyl acetate, and then gradually increasing the polarity by adding methanol by 2% increment up to 80% methanol. Fractions 20–25 afforded boldine metabolite **2** (nor-boldine) which was soluble in chloroform (5 mg, yield = 1.25%, R*f* = 0.75, S1), in the form of white amorphous powder. Fractions 45–50 afforded 3 (5 mg, 1.25% yield, R*f* = 0.3, S1), as amorphous white powder.

#### 3.4.3. Boldine Transformation by *Penicillium chrysogeneum* ATCC 10002

Boldine was dissolved in DMF (100 mg/2.5 mL) and was equally partitioned among 10 stage II flasks. After 15 days, 200 mg fermentation residue was obtained. The residue was dissolved in 0.5 mL of a methanol–dichloromethane mixture (1:1) and placed on top of silica gel column (30 g, 100 cm × 1.5 cm). The column was eluted by the gradient elution method using 100% ethyl acetate, and then gradually increasing the polarity by adding methanol by 5% increment up to 80% methanol. Fractions 72–83 afforded impure boldine metabolite **4**. Those fractions were combined and evaporated to dryness (10 mg) Further purification was achieved utilizing the Sephadex LH-20 column (5 g, 60 cm × 1 cm), eluted severally with 100% methanol to yield pure boldine metabolite **4** (boldine-O-ß-D-glucoside, 4 mg (4% yield, *Rf* = 0.13, S1), as amorphous white powder.

#### 3.4.4. Berberine Transformation by *Cunninghamella blackesleeana* NRRL 1369

Berberine (200 mg/5 mL sterile distilled water with gentle heating) was equally distributed among 20 stage II flasks. After 14 days, fermentation residue (300 mg) was dissolved in 3 mL of methanol and placed on top of Sephadex column (30 g, 120 × 2.5 cm^2^) eluted with methanol. In this case, 40 fractions (10 mL each) were collected. Pure berberine metabolite **5** (tridemethyl berberine) was obtained as amorphous white powder (6 mg, 3% yield, R*f* = 0.2, S2), from fractions (20–28).

#### 3.4.5. Berberine Transformation by *Penicillium chrysogeneum* ATCC 10002

Berberine (200 mg/5 mL sterile distilled water with gentle heating) was partitioned equally among 20 stage II flasks. 320 mg fermentation residue was obtained after 10 days. The residue was dissolved in 3 mL of methanol and placed on top of a glass column (30 g, 120 × 2.5 cm^2^) packed with slurry of Sephadex LH-20 in methanol. In this case, 30 fractions (10 mL each) were collected. Fractions (10–17) yielded a pure berberine metabolite **6** (demethylene berberine) as amorphous white powder (5 mg, 2.5% yield).

#### 3.4.6. Berberine Transformation by *Cunninghamella elegans* NRRL 2310

Berberine (200 mg/5 mL sterile distilled water with gentle heating) was distributed equally among 20 stage II flasks. Fermentation residue (320 mg) was obtained after 13 days, and dissolved in 3 mL of methanol and placed onto Sephadex LH-20column (30 g, 120 × 2.5 cm^2^) eluted with methanol. In this case, 40 fractions (10 mL each) were collected Fractions (25–32) yielded a pure berberine metabolite **7** (dihydroberberine) as amorphous white powder (5 mg, 2.5% yield).

### 3.5. Molecular Docking

Molegro Virtual docker 2019 7.0.0 (MVD, a full trial version) was used to perform the docking studies. The crystal structures of the template proteins, PDB: 6E53 and 2F70, were downloaded from the Protein Data Bank. The crystal structure 6E53 was obtained by X-ray diffraction at a resolution of 2.8 Å and represents telomerase enzyme (TERT) from *Tribolium castaneum* in complex with MeCITP as an inhibitor [31]. The crystal structure 2F70 was obtained by X-ray diffraction at a resolution of 2.12 Å and is for tyrosine phosphatase 1B (PTP-1B) in complex with 1,2,3,4-tetrahydroisoquinolyl sulfamic acid derivative as an inhibitor [35]. MervinSketch software (ChemAxon^®^ suite) was used to acquire the 3D models of the structures from which the lowest energy conformer for each compound was saved as structure data file (.sdf) format for docking. For each docking study, the protein structure and the ligand molecules were imported into the workspace of MVD. The total energy of the protein complexes was inspected using Ligand energy inspector of MVD software and the individual energy for different types of binding were analyzed. The parameters used were the same as previously reported in our studies [36,37]. The docking cavities constraints were set to give the best redocking pose. For redocking to TERT the selected constraints were X = 3.55, Y = −11.66, Z = 113.17 and radius = 8 Å, while for PTP-1B were X = −14.58, Y = 46.76, Z = 49.48 and radius = 10 Å. Redocking of the inhibitors co-crystalized with the proteins under study was performed to validate the docking parameters. The best redocked pose of 5-MeCITP had RMSD of 1.02 Å while for 1,2,3,4-tetrahydroisoquinolyl sulfamic acid derivative had RMSD of 0.84 Å. The aforementioned constraints were applied to the docking of the test compounds of boldine and berberine metabolites, respectively. The scoring function was set to MolDock score (GRID) and Grid resolution of 0.30 Å, while the search algorithm was set to MolDock Simplex Evolution. After running the docking simulation, the results were examined for determining the docking scores and the interactions with the amino acid residues at the docking cavity.

## Figures and Tables

**Figure 1 pharmaceuticals-15-01195-f001:**
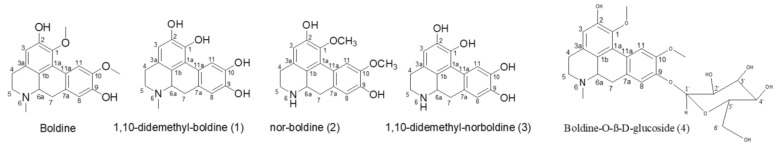
Structure of boldine and its metabolites.

**Figure 2 pharmaceuticals-15-01195-f002:**
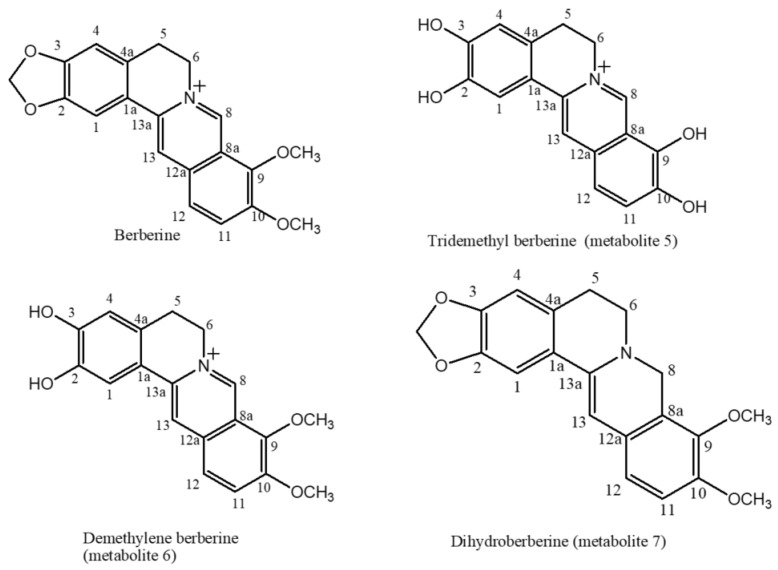
Structure of berberine and its metabolites.

**Figure 3 pharmaceuticals-15-01195-f003:**
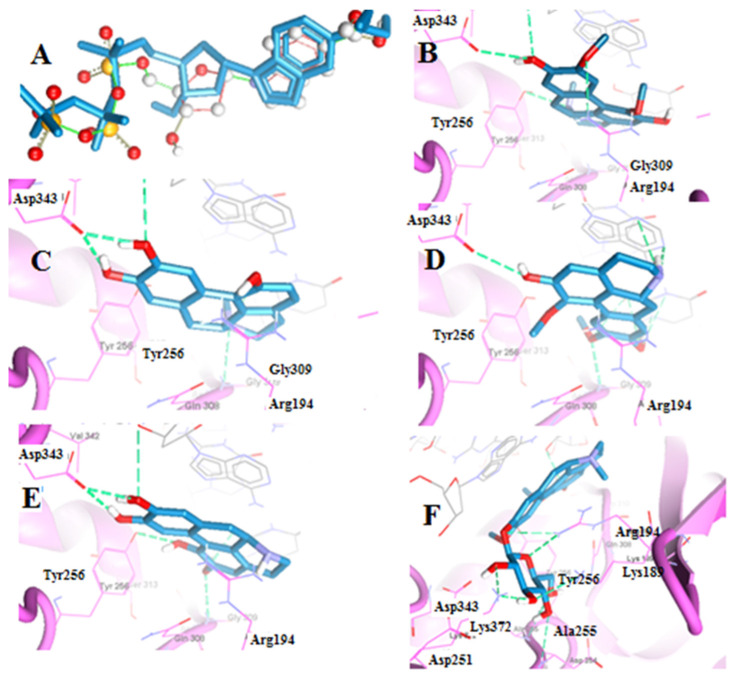
(**A**) Superimposed co-crystalized and redocked poses of 5-MeCITP in TERT. (**B**–**F**) Amino acid residues involved in ligand interaction shown as dashed lines with boldine, boldine metabolites 1, 2, 3 and 4, respectively.

**Figure 4 pharmaceuticals-15-01195-f004:**
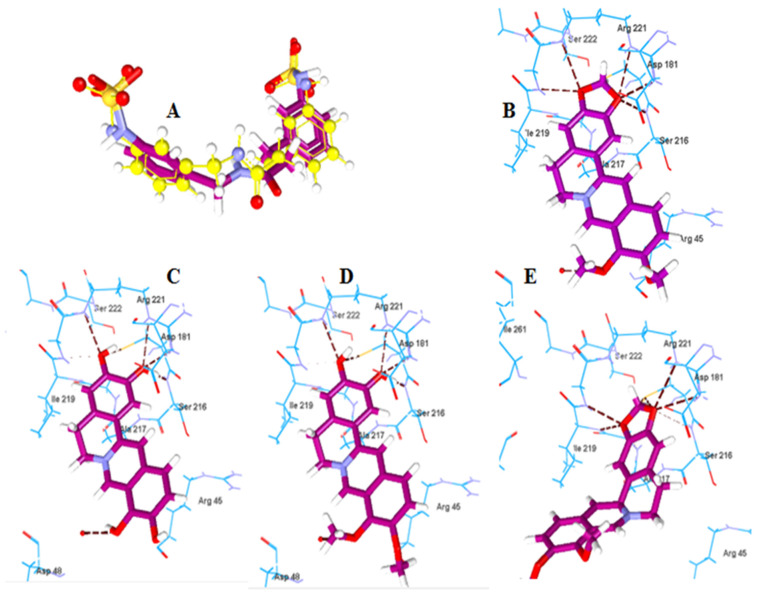
(**A**) Superimposed co-crystalized and redocked poses of 2F70 in PTP-1B. (**B**–**E**) Amino acid residues involved in ligand interaction shown as dashed lines with berberine, berberine metabolites **5**, **6**, and **7**, respectively.

**Table 1 pharmaceuticals-15-01195-t001:** Results of ^1^H NMR and ^13^C NMR analysis of boldine metabolites **1**-**3** and boldine (in CDCl_3_, 400 MHz) and boldine metabolite **4** (in Acetone-*d_6_*).

Position	δ_H_, Integration, Multiplicities (J in Hz)					δ_C_, Multiplicities				
	Boldine	Boldine Metabolite-1	Boldine Metabolite-2	Boldine Metabolite-3	Boldine Metabolite-4	Boldine	Boldine Metabolite-1	Boldine Metabolite-2	Boldine Metabolite-3	Boldine Metabolite-4
1	---	---	---	---	---	142.6, C	138.3, C	143.3, C	139.3, C	141.8, C
1a	---	---	---	---	---	122.8, C	125.3, C	125.3, C	123.6, C	126.1, C
1b	---	---	---	---	---	126.2, C	123.8, C	126.2, C	125.3, C	125.0, C
2	---	---	---	---	---	149.1, C	149.3, C	149.0, C	147.5, C	148.3, C
3	6.49, 1H, s	6.52, 1H, s	6.64, 1H, s	6.48, 1H, s	6.52, 1H, s	114.1, CH	114.6, CH	115.0, CH	112.2, CH	114.2, CH
3a	---	---	---	---	---	128.8, C	128.2, C	128.2, C	128.6, C	130.0, C
4	2.24, 2H, m	2.52, 2H, br	2.52, 2H, br	2.77, 2H, br	2.51, 2H, m	28.5, CH_2_	28.1, CH_2_	28.9, CH_2_	26.3, CH_2_	28.4, CH_2_
5	2.50, 2H, m	2.97, 2H, br	3.02, 2H, br	2.94, 2H, br	2.96, 2H, m	52.8, CH_2_	53.4, CH_2_	42.4, CH_2_	45.1, CH_2_	52.4, CH_2_
6a	2.73, 1H, dd (3.72, 13.8)	2.66, 1H, m	2.65, 1H, m	2.90, 1H, m	2.65, 1H, m	50.3, CH	60.8, CH	52.7, CH	55.3, CH	61.3, CH
7	2.83, 2H, m	3.15, 2H, m	3.20, 2H, m	3.11, 2H, m	3.20, 2H, m	33.8, CH_2_	32.3, CH_2_	34.7, CH_2_	35.1, CH_2_	33.3, CH_2_
7a	---	---	---	---	---	129.6, C	128.6, C	130.6, C	130.4, C	130.6, C
8	6.71, 1H, s	6.68, 1H, s	6.74, 1H, s	6.68, 1H, s	6.68, 1H, s	115.2, CH	115.4, CH	114.9, CH	115.7, CH	118.4, CH
9	---	---	---	---	---	146.0, C	144.5, C	145.1, C	147.3, C	146.2, C
10	---	---	---	---	---	145.8, C	142.5, C	145.7, C	140.2, C	149.3, C
11	7.85, 1H, s	7.87, 1H, s	7.89, 1H, s	7.73, 1H, s	7.87, 1H, s	111.9, CH	110.3, CH	112.2, CH	111.2, CH	110.4, CH
11a	---	---	---	---	---	125.6, C	120.2, C	123.2, C	122.5, C	127.3, C
1-OCH_3_	3.55, 3H, s	---	3.58, 3H, s	---	3.56, 3H, s	59.2, CH_3_	---	58.4, CH_3_	---	59.1, CH_3_
10-OCH_3_	3.77, 3H, s	---	3.76, 3H, s	---	3.77, 3H, s	55.7, CH_3_	---	55.3, CH_3_	---	55.2, CH_3_
N-CH_3_	2.38, 3H, s	2.38, 3H, s	---	---	2.37, 3H, s	43.7, CH_3_	44.3, CH_3_	---	---	44.2, CH_3_
Glc-1`	---	---	---	---	5.05, 1H, d (7.2)	---	---	---	---	101.2, CH
Glc-2`	---	---	---	---	4.00–4.26, 6H, m	---	---	---	---	77.2, CH
Glc-3`	---	---	---	---		---	---	---	---	77.0, CH
Glc-4`	---	---	---	---		---	---	---	---	73.9, CH
Glc-5`	---	---	---	---		---	---	---	---	70.4, CH
Glc-6`	---	---	---	---		---	---	---	---	61.8, CH_2_

**Table 2 pharmaceuticals-15-01195-t002:** Results of ^1^H NMR and ^13^C NMR analysis of berberine metabolites **5**–**7** and berberine [27] (in MeOD, 400 MHz).

Position	δ_H_, Integration, Multiplicities (J in Hz)				δ_C_, Multiplicities			
	Berberine	Berberine Metabolite-5	Berberine Metabolite-6	Berberine Metabolite-7	Berberine	Berberine Metabolite-5	Berberine Metabolite-6	Berberine Metabolite-7
1	7.67, 1H, s	7.82, 1H, s	7.57, 1H, s	6.73, 1H, s	105.4, CH	104.7, CH	105.8, CH	106.7, CH
1a	---	---	---	---	120.4, C	122.5, C	120.5, C	122.3, C
2	---	---	---	---	147.6, C	148.2, C	146.9, C	145.9, C
3	---	---	---	---	149.7, C	150.3, C	149.5, C	148.3, C
4	6.97, 1H, s	7.07, 1H, s	6.87, 1H, s	6.58, 1H, s	108.4, CH	108.0, CH	109.0, CH	110.5, CH
4a	---	---	---	---	130.6, C	131.7, C	128.0, C	129.3, C
5	3.27, 2 H, t (4)	3.21, 2H, t (4)	3.12, 2H, t (4)	2.91, 2H, t (4)	26.4, CH_2_	27.0, CH_2_	25.9, CH_2_	28.4, CH_2_
6	4.94, 2 H, t (4)	4.94, 2H, t (4)	4.89, 2H, t (4)	3.21, 2H, t (4)	55.2, CH_2_	54.1, CH_2_	55.0, CH_2_	53.1, CH_2_
8	9.77, 1H, s	9.87, 1H, s	9.82, 1H, s	4.30, 2H, s	145.4, CH	143.9, CH	144.3, CH	51.4, CH_2_
8a	---	---	---	---	121.4, C	122.4, C	120.9, C	120.7, C
9	---	---	---	---	143.6, C	145.4, C	144.3, C	145.2, C
10	---	---	---	---	150.4, C	151.4, C	150.3, C	149.2, C
11	8.12, 1H, d (9)	8.33, 1H, d (8)	8.18, 1H, d (8)	7.18, 1H, d (8)	126.7, CH	125.9, CH	126.2, CH	125.0, CH
12	8.01, 1H, d (9)	7.92, 1H, d (8)	7.99, 1H, d (8)	7.22, 1H, d (8)	123.5, CH	123.2, CH	123.7, CH	121.5, CH
12a	---	---	---	---	132.9, C	133.0, C	133.0, C	130.5, C
13	8.71, 1H, s	8.88, 1H, s	8.75, 1H, s	5.93, 1H, s	120.2, CH	121.4, CH	119.8, CH	100.8, CH
13a	---	---	---	---	137.4, C	136.8, C	136.7, C	138.4, C
O-CH_2_-O	6.11, 2H, s	---	---	5.90, 2H, s	102.1, CH_2_	---	---	101.2, CH_2_
O-9-CH_3_	4.21, 3H, s	---	4.10, 3H, s	4.20, 3H, s	62.0, CH_3_	---	61.4, CH_3_	61.3, CH_3_
O-10-CH_3_	4.11, 3H, s	---	4.07, 3H, s	3.80, 3H, s	57.1, CH_3_	---	56.5, CH_3_	56.2, CH_3_

**Table 3 pharmaceuticals-15-01195-t003:** Docking scores and the amino acids involved in hydrogen bonding interactions.

Compound	TERT		Compound	PTP-1B	
	MolDock Score	Interacting Amino Acid		MolDock Score	Interacting Amino Acid
Boldine	−120.14	Arg194, Asp343, Tyr256	Berberine	−123.57	Cys215, Ser216, Gly220, Arg221
Boldine metabolite **1**	−112.52	Asp343, Gly309	Berberine metabolite 5	−115.91	Cys215, Ser216, Arg221
Boldine metabolite **2**	−135.54	Asp343, Gly309	Berberine metabolite 6	−115.60	Cys215, Ser216, Arg221
Boldine metabolite **3**	−119.00	Asp343, Gly309, Tyr256	Berberine metabolite 7	−106.05	Cys215, Ile219, Gly220, Arg221
Boldine metabolite **4**	−151.75	Arg194, Asp254, Ala255, Lys372, Lys189			

## Data Availability

Data is contained within the article and Appendix A.

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
