# Peer review of "Biotransformation of Modified Benzylisoquinoline Alkaloids: Boldine and Berberine and In Silico Molecular Docking Studies of Metabolites on Telomerase and Human Protein Tyrosine Phosphatase 1B"

_pharmaceuticals, 2022, doi:10.3390/ph15101195_

Round 1

Reviewer 1 Report

The manuscript “Biotransformation of Modified Benzylisoquinoline Alkaloids: Boldine and Berberine and In Silico Molecular Docking Studies of Metabolites on Telomerase and Human Protein Tyrosine Phosphatase 1B” by Eliwa et al. presents the study on biotransformation of boldine and berberine to their metabolites using selected filamentous fungi. Furthermore, the authors isolated and characterized several metabolites of boldine and berberine and using in silico approach they tested their potential to bind/inhibit telomerase and protein tyrosine phosphatase 1B, respectively. In my view this study is interesting and brings novel findings on biotransformation of two important natural agents.

However, some minor corrections are needed before publication. My comments and recommendations are outlined below:

Lines 42-43: Should read “This process is achieved by microorganisms such as bacteria and fungi, or by activities of various enzymes.”

Lines 49-50: Should read “Boldine is a free radical scavenger implicated in the treatment of various diseases.”

Lines 52-53: Should read “Furthermore, boldine has gastroprotective effect and is useful in gastric ulcer treatment.”

Lines 53-54: It would be better stating that “As an effective antioxidant, boldine can act as anti-apoptotic agent and protect the liver from damage induced by methylprednisolone.”

Line 70: The anticancer activity of berberine should be introduced here. It is known that berberine and its derivatives are potent antiproliferative and apoptosis inducing agents.

Table 1: Unify the table. There are small and capitals used for metabolites.

Lines 100 - onward: Please, unify the terminology for the studied metabolites within the manuscript. I recommend sticking to “boldine metabolite X” and “berberine metabolite X.

Line 186 vs. Figure 2: Which one is correct? Tridesmethylberberine (line 186) or tridemethyl berberine (Figure 2).

Figure 2: Metabolite 7 should be “dihydroberberine”.

Lines 222-225: Reformulate this sentence. It does not make sense.

Line 241: The authors have to discuss here in more detail the bioavailability and health benefits of isolated metabolites compare to the absorption/activity of berberine or boldine, including the improvement/increase or decrease of their therapeutic potential. Relevant references should be cited.

Line 253: Should read “…  boldine, boldine metabolites 1, 2, 3 and 4 were …”

Lines 286-287: Should read “Berberine and its metabolites (5-7) were docked to PTP-1B enzyme to predict their antidiabetic potential.”

Lines 361 and 364: Should read “Fractions 50–66 afforded impure boldine metabolite 1.”,  and “… to yield pure boldine metabolite 1 (1,10-didemethyl-boldine), …”

Lines 374 and 376: Should read “Fractions 20–25 afforded boldine metabolite 2 (nor-boldine) which …”, and “Fractions 45–50 afforded boldine metabolite 3 (1,10-didemethyl-norboldine) …”

Lines 386 and 389: Should read “Fractions 72–83 afforded impure boldine metabolite 4.”, and “… pure boldine metabolite 4 (boldine-O-ß-D-glucoside, 4 mg, …”

Lines 392, 401 and 410:  Berberine is sparingly soluble in water and aqueous buffers. Was the berberine really dissolved in water?

Line 398: Should read “Pure berberine metabolite 5 (tridemethyl berberine) was obtained …”

Line 407: Should read “Fractions (10–17) yielded a pure berberine metabolite 6 (demethylene berberine) …”

Line 416: Should read “Fractions (25–32) yielded a pure berberine metabolite 7 (dihydroberberine) …”

Lines 445-467, and Supplementary materials: The authors should consider to unify the manuscript and use terms as “boldine metabolite X” and “berberine metabolite X”, instead of terms “metabolite-1-7” or number “7” in Figure S40.

Author Response

Reviewer 1: The manuscript “Biotransformation of Modified Benzylisoquinoline Alkaloids: Boldine and Berberine and In Silico Molecular Docking Studies of Metabolites on Telomerase and Human Protein Tyrosine Phosphatase 1B” by Eliwa et al. presents the study on biotransformation of boldine and berberine to their metabolites using selected filamentous fungi. Furthermore, the authors isolated and characterized several metabolites of boldine and berberine and using in silico approach they tested their potential to bind/inhibit telomerase and protein tyrosine phosphatase 1B, respectively. In my view this study is interesting and brings novel findings on biotransformation of two important natural agents.

Our reply: Thank you for your precious time and your valuable comments

However, some minor corrections are needed before publication. My comments and recommendations are outlined below:

Lines 42-43: Should read “This process is achieved by microorganisms such as bacteria and fungi, or by activities of various enzymes.”

Reply: corrected as required and marked up using the “Track Changes” function

Lines 49-50: Should read “Boldine is a free radical scavenger implicated in the treatment of various diseases.”

Reply: corrected as required and marked up using the “Track Changes” function

Lines 52-53: Should read “Furthermore, boldine has gastroprotective effect and is useful in gastric ulcer treatment.”

Reply: corrected as required and marked up using the “Track Changes” function

Lines 53-54: It would be better stating that “As an effective antioxidant, boldine can act as anti-apoptotic agent and protect the liver from damage induced by methylprednisolone.”

Reply: corrected as required and marked up using the “Track Changes” function

Line 70: The anticancer activity of berberine should be introduced here. It is known that berberine and its derivatives are potent antiproliferative and apoptosis inducing agents.

Reply: Many thanks for your suggestion. The required information and the relevant references were added, Wang, Y.; Liu, Y.; Du, X.; Ma, H.; Yao, J., The Anti-Cancer Mechanisms of Berberine: A Review. Cancer Manag Res. 2020, 30, (12), 695-702. Changes are marked up using the “Track Changes” function

Table 1: Unify the table. There are small and capitals used for metabolites.

Reply: corrected as required and marked up using the “Track Changes” function

Lines 100 - onward: Please, unify the terminology for the studied metabolites within the manuscript. I recommend sticking to “boldine metabolite X” and “berberine metabolite X.

Reply: corrected as required and marked up using the “Track Changes” function

Line 186 vs. Figure 2: Which one is correct? Tridesmethylberberine (line 186) or tridemethyl berberine (Figure 2).

Reply: Corrected, tridemethyl berberine is the correct one

Figure 2: Metabolite 7 should be “dihydroberberine”.

Reply: corrected as required and marked up using the “Track Changes” function

Lines 222-225: Reformulate this sentence. It does not make sense.

Reply: The sentences was formulated as requested and marked up using the “Track Changes” function and highlighted in yellow color.

The new sentence is (Dealkylation is the most common reaction occurred in boldine and berberine indicating that several monooxygenases with different substrate specificities are involved in this reaction.

Line 241: The authors have to discuss here in more detail the bioavailability and health benefits of isolated metabolites compare to the absorption/activity of berberine or boldine, including the improvement/increase or decrease of their therapeutic potential. Relevant references should be cited.

Reply: The required information was discussed as follows: It was found that the demethylation of berberine enhances its intestinal absorption in mice in vivo, which increases the therapeutic potential of berberine [28]. Furthermore, dihydroberberine is considered an active metabolite as it showed up to five times more bioavailable/absorbable and two times longer-lasting than berberine (8 hours vs. 4 hours) [29].

The sentence is marked up using the “Track Changes” function and highlighted in yellow color.

[28] Zhang, Z.; Cong, L.; Peng, R.; Han, P.; Ma, S. R.; Pan, L. B.; Fu, J.; Yu, H.; Wang, Y.; Jiang, J. D., Transformation of berberine to its demethylated metabolites by the CYP51 enzyme in the gut microbiota. Journal of pharmaceutical analysis 2021. 11, (5), 628–637.

[29] Moon, JM.; Ratliff, KM.; Hagele, AM.; Stecker, RA.; Mumford, PW.; Kerksick, CM., Absorption Kinetics of Berberine and Dihydroberberine and Their Impact on Glycemia: A Randomized, Controlled, Crossover Pilot Trial. Nutrients. 2021. 14, (1), 124.

 Line 253: Should read “…  boldine, boldine metabolites 1, 2, 3 and 4 were …”

Reply: corrected as required and marked up using the “Track Changes” function

Lines 286-287: Should read “Berberine and its metabolites (5-7) were docked to PTP-1B enzyme to predict their antidiabetic potential.”

Reply: corrected as required and marked up using the “Track Changes” function

Lines 361 and 364: Should read “Fractions 50–66 afforded impure boldine metabolite 1.”,  and “… to yield pure boldine metabolite 1 (1,10-didemethyl-boldine), …”

Reply: corrected as required and marked up using the “Track Changes” function

Lines 374 and 376: Should read “Fractions 20–25 afforded boldine metabolite 2 (nor-boldine) which …”, and “Fractions 45–50 afforded boldine metabolite 3 (1,10-didemethyl-norboldine) …”

Reply: corrected as required and marked up using the “Track Changes” function

Lines 386 and 389: Should read “Fractions 72–83 afforded impure boldine metabolite 4.”, and “… pure boldine metabolite 4 (boldine-O-ß-D-glucoside, 4 mg, …”

Reply: corrected as required and marked up using the “Track Changes” function

Lines 392, 401 and 410:  Berberine is sparingly soluble in water and aqueous buffers. Was the berberine really dissolved in water?

Reply: Yes, with gentle heating

Line 398: Should read “Pure berberine metabolite 5 (tridemethyl berberine) was obtained …”

Reply: corrected as required and marked up using the “Track Changes” function

Line 407: Should read “Fractions (10–17) yielded a pure berberine metabolite 6 (demethylene berberine) …”

Reply: corrected as required and marked up using the “Track Changes” function

Line 416: Should read “Fractions (25–32) yielded a pure berberine metabolite 7 (dihydroberberine) …”

Reply: corrected as required and marked up using the “Track Changes” function

Lines 445-467, and Supplementary materials: The authors should consider to unify the manuscript and use terms as “boldine metabolite X” and “berberine metabolite X”, instead of terms “metabolite-1-7” or number “7” in Figure S40.

Reply: corrected as required and marked up using the “Track Changes” function

Reviewer 2 Report

This is an interesting manuscript dealing with Biotransformation of Modified Benzylisoquinoline Alkaloids.  Minor revision is needed to tell readers how to relate the Molecular docking  scores with the biofunctionality.  As can be seen from Table 3, for Bolding and 4 derivatives, compound 4 is best as it has 'largest' score while for Berberine and 3 derivatives, compound 7 is best as it has lowest score.  It should be accepted after minor revision.

Author Response

Reviewer 2:

This is an interesting manuscript dealing with Biotransformation of Modified Benzylisoquinoline Alkaloids.  Minor revision is needed to tell readers how to relate the Molecular docking scores with the biofunctionality.  As can be seen from Table 3, for Bolding and 4 derivatives, compound 4 is best as it has 'largest' score while for Berberine and 3 derivatives, compound 7 is best as it has lowest score.  It should be accepted after minor revision.

Our Reply: Thank you for your precious time and your valuable comments

Relation of the Molecular docking scores with the biofunctionalities was discussed in sections 1.2.1. and 1.2.2. and highlighted in yellow color.
